# Variability in Minimal-Damage Sap Flow Observations and Whole-Tree Transpiration Estimates in a Coniferous Forest

**Junjun Yang [1,2], Zhibin He [2,\*], Pengfei Lin [2], Jun Du [2], Quanyan Tian [2], Jianmin Feng [1], Yufeng Liu [1], Lingxia Guo [1], Guohua Wang [3,4], Jialiang Yan [5] and Weijun Zhao [6]**

1. College of Geography & Environment, Xianyang Normal University, Xianyang 712000, China
2. Linze Inland River Basin Research Station, Chinese Ecosystem Research Network, Key Laboratory of Eco-Hydrology of Inland River Basin, Cold and Arid Regions Environmental and Engineering Research Institute, Chinese Academy of Sciences, Lanzhou 730000, China
3. College of Geographical Sciences, Shanxi Normal University, Taiyuan 030000, China
4. Key Laboratory of Desert and Desertification, Northwest Institute of Eco-Environment and Resources, Chinese Academy of Sciences, Lanzhou 730000, China
5. Institute of Geography Science, Taiyuan Normal University, Jinzhong 030619, China
6. Academy of Water Resources Conservation Forests in Qilian Mountains of Gansu Province, Zhangye 734000, China
\* Correspondence: hzbmail@lzb.ac.cn; Tel.: +86-093-1496-7165

**Abstract:** Transpiration is fundamental to the understanding of the ecophysiology of planted forests in arid ecosystems, and it is one of the most uncertain components in the ecosystem water balance. The objective of this study was to quantify differences in whole-tree transpiration estimates obtained with a heat ratio probe in a secondary Qinghai spruce (*Picea crassifolia*) forest. To do this, we analyzed the sap flux density values obtained with sensors installed in (1) holes drilled in the preceding growing season (treatment) and (2) holes drilled in the current year (control). The study was conducted in a catchment in the Qilian Mountains of western China. The results showed that an incomplete diameter at breast height (DBH) range contributed to 28.5% of the overestimation of the sapwood area when the DBH > 10 cm and 22.6% of the underestimation of the sapwood area when the DBH < 5 cm. At daily scales, there were significant differences in both the quantity and magnitude of the sap flux density between the treatment and control groups. Furthermore, a linear regression function ($R^2$ = 0.96, $p$ < 0.001), which was almost parallel to the 1:1 reference line, was obtained for the sap flux density correction for the treatment group, and the daily sap flux density and whole-tree transpiration were underestimated by 36.8 and 37.5%, respectively, at the half-hour scale. This study illustrates uncertainties and a correction function for sap flow estimations in young Qinghai spruce trees when using heat ratio sensors with minimal damage over multiple growing seasons.

**Keywords:** Qinghai spruce; sap flow; transpiration; heat ratio method (HRM); upper Heihe River Basin

## 1. Introduction

Transpiration is the largest type of water flux, making up 80 to 90 percent of terrestrial evapotranspiration [1]. However, transpiration is also an important source of uncertainty in the land surface modeling of global hydrological cycles and water balance quantification [2,3]. The differences in estimates arise from differences in the methods used to determine transpiration at the tree and stand levels in forested landscapes [4]. Accurate estimations of whole-tree and stand-level water use will improve our ability to constrain land surface models [5].

Transpiration measurements have a long history and have been obtained in a variety of ways [6–8]. Marshall [9] measured the rate of sap flow in conifers using the principles of heat transport to estimate the tree transpiration. McNaughton and Black [10] used the energy balance to measure the evapotranspiration from a young Douglas fir forest.

Edwards and Warwick [11] compared heat pulse velocity and Penman-Monteith equation methods and demonstrated that the heat pulse velocity method is promising for measuring transpiration rates. On that basis, Barrett et al. [12] determined the thresholds of sap velocity for *Eucalyptus maculate*, *Ceratopetalum apetalum*, and *Doryphora sassafras* to identify a range of sap flux velocities that could be accurately estimated from the heat pulse velocity technique. Currently, the heat ratio method (HRM) is accepted as an important improvement for measurements of low and reverse rates of sap flow in woody plants [13].

The HRM is inexpensive, easy to implement, has a reasonable time resolution, and can be used with minimal disruption and damage to the sap stream [5,14]. It is also one of the most accurate sap flow measurement techniques [8]. However, the HRM has some drawbacks. For example, the HRM relies on empirical calibrations, and variability in the physiology of the sampling trees results in high sensitivity to inaccurate probe spacing [13]. Green et al. [14] reported that the size and geometry of sensors contributed to sap flow disturbances, and the wound width had a large influence on the measurements. Further, repeated drilling inevitably causes damage to observation trees used over multiple growing seasons for sap flow observations (Figure 1), with young trees being more affected than older trees [15]. To minimize the observation damage and to protect young trees, original sensor holes may be reused. Here, we focus on exploring the uncertainty and variability in sap flow observations for transpiration estimates.

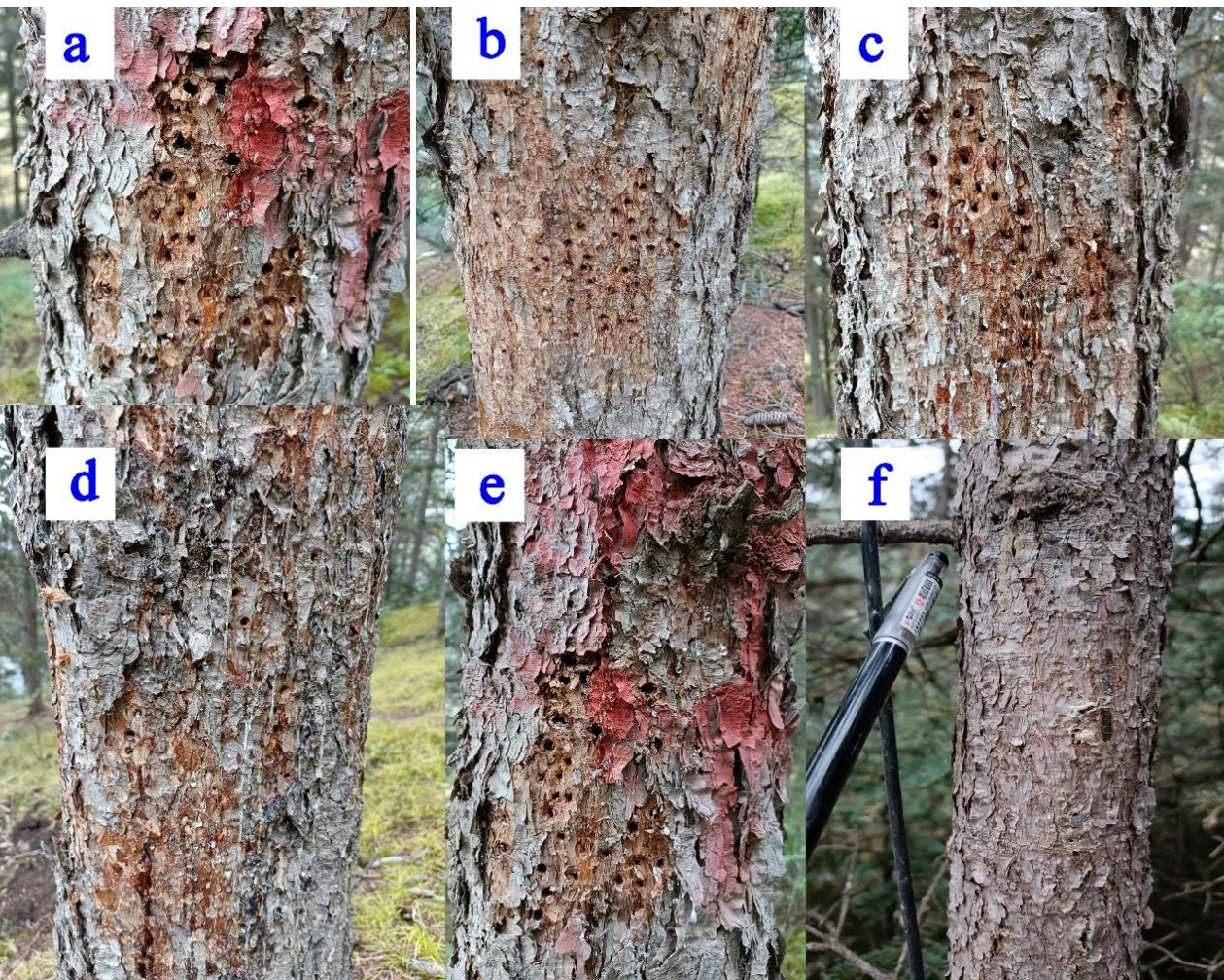

**Figure 1.** Tree wounds after several growing season observations in Qinghai spruce in the Qilian Mountains. (**a**–**f**) were six sample tress after multiple observation seasons.

The Qilian Mountains are the source of the Heihe River and its basin, the second-largest inland river basin in China. Fifty percent of the Qilian Mountains area is occupied by forest [16], with Qinghai spruce (*Picea crassifolia*) accounting for 25% of the total forest area [17]. Accurate estimations of forest transpiration are important for understanding the components of water balance at the stand, catchment, and regional scales, and for the ecohydrological function of forests in this arid region. To quantify the forest transpiration, we measured the sap flow using HRM sensors in eight healthy trees over multiple growing seasons in the Pailugou Catchment. Our objectives were to (1) increase the accuracy of sap flux estimates, especially for young trees; (2) examine the variations in sap flux density in treatment and control groups over two consecutive growing seasons; and (3) evaluate the relative error and determine the correction function for the Qinghai spruce sap flow estimates.

## 2. Materials and Methods

### 2.1. Site Description

The study was performed in the Qilian Mountains (100°17′ E, 38°24′ N), located upstream of the Heihe River Basin, approximately 50 km south of Zhangye, Gansu, China. The annual rainfall in the area ranges between 266.3 and 471.2 mm (data for 1994 to 2015) at an elevation of 2700 m. Almost 65% of the precipitation events occur during the summer (from July to September). The pan evaporation is about 1129.8 mm per year. The mean annual air temperature is 0.5 °C, with the lowest and highest temperatures being −36.0 and 28.0 °C, respectively. The mean annual vapor pressure deficit (VPD) is 0.31 kPa, with a growing season (between July and September) VPD of 0.43 kPa. The annual frost-free period is about 165 days.

The vegetation in the study area is dominated by a secondary forest of Qinghai spruce, ranging in age from 15 to 120 years old, and a mean stand density of 2418 trees ha$^{-1}$. The average tree height was 11.8 ± 2.8 m and average diameter at breast height (DBH) was 9.77 ± 7.87 cm (determined between July and September of 2013 from 21 sample plots of 30 × 30 m$^2$), at an altitude gradient range of 2700 to 3200 m (Figure 2). Moss (*Abietinella abietina*) covered 95% of the forest floor to a thickness of 13.7 ± 4.5 cm. The soils are gray-drab, with a bulk density of 0.95 g cm$^{-3}$ and an average soil depth to bedrock of 0.7 m.

### 2.2. Meteorology and Soil Moisture Measurements

The meteorological data (e.g., daily average value) were obtained from two weather stations. One station was an Environmental Integration System (ENVIS, IMKO Micro modul technik GmbH, Ettlingen, Germany), located at 100 m distance from the forest boundary; the rainfall rate and intensity were continuously recorded at 1- and 30-min intervals, respectively. The other station was a conventional ground meteorological observation station for long-term observations, about 50 m away from the ENVIS. The meteorological variables, such as air temperature, relative humidity, sunshine duration, wind speed, atmospheric pressure, surface temperature, pan evaporation, and depth of soil freezing, were manually recorded at 8:00, 14:00, and 20:00 daily. Because the sap flow is sensitive to VPD, the global short-wave radiation, air temperature, and soil moisture [18], weather variability, and layered soil water content were recorded during the 2014 and 2015 growing seasons at 30-min intervals (Figure 2).

The soil water content was continuously monitored with an EM50 data logger (Decagon Devices, Inc. 2365 NE Hopkins Ct. Pullman, WA 99163, USA) at three locations with 5TM sensors placed at 5, 10, 20, 40, and 60 cm depths within the sample plot; before application, the records were calibrated against the gravimetric method. The volumetric soil moisture of the Qinghai spruce forest ranged from 0.13 to 0.27 m$^3$ m$^{-3}$, with a mean value of 0.18 ± 0.03 m$^3$ m$^{-3}$ during the 2015 growing season.

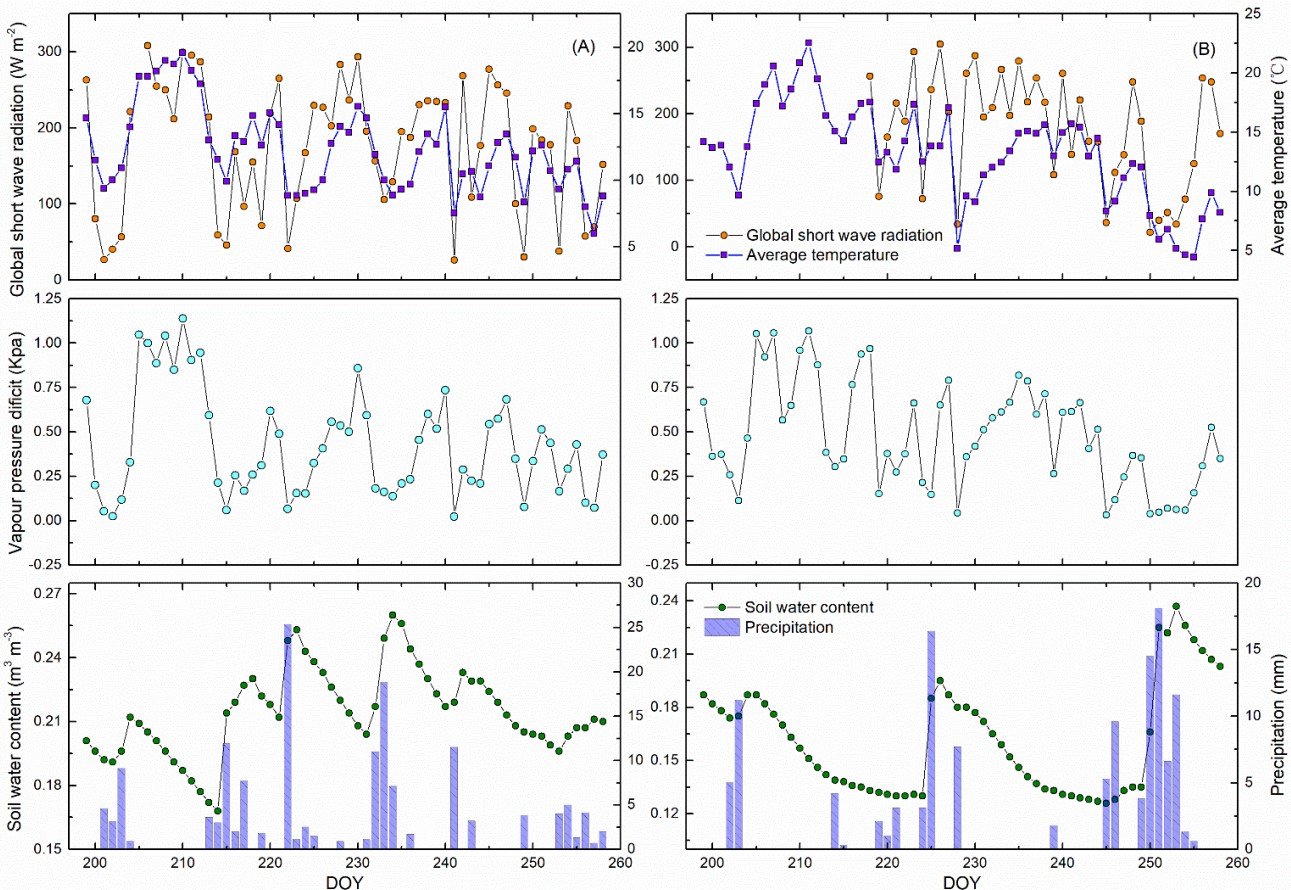

**Figure 2.** The environmental conditions during the growing seasons in the treatment group (figures in column **A**) and control group (figures in column **B**).

### 2.3. Heat Ratio Method and Probe Installation

　　The HRM is one of the most widely used methods for determining sap flow rates [19]. By measuring the ratio of heat transported to two symmetrically placed temperature sensors, the magnitude and direction of water flux can be calculated [13]. The HRM30 sensors (SFM1 Sap Flow Meters ICT International PTY Ltd., Armidale, NSW, Australia) can be used on lignified tissue > 10 mm in diameter. Two thermocouple junctions (located at 12.5 and 27.5 mm from the epoxy base of the temperature-sensing probe) fitted on two temperature probes (35 mm length) can give more accurate measurements than one thermocouple junction. The heater probe is located in the middle, between the downstream and upstream probes. Each SFM was configured with an external 12 V 35AH battery (accepted voltage 7 to 28 V). The records from each SFM were logged with an SL5 smart logger and downloaded to a computer each month.

　　Eight sets of HRM30 sensors were installed in eight individual trees with a mean DBH similar to the average DBH of the trees in the experiment plot. The tree parameters (e.g., crown width, diameter at breast height, sapwood width, and height of tree) of the sampling trees were measured before sensor installation (Table 1). Then, we selected a placement site with a flat and straight stem section and removed the bark at breast height (50 cm height for the trees, with DBH < 10 cm) to expose the cambium.

**Table 1.** Biometric and physiological parameters of the sap flow measurement trees.

| No. | Groups of DBH | Height of Tree (m) | Crown Width (m) | Diameter at Breast Height (cm) | Depth of Bar (cm) | Sapwood Width (cm) | Sapwood Area (cm$^2$) | Canopy's Projected Area (m$^2$) |
|---|---|---|---|---|---|---|---|---|
| #1 | DBH1 | 16.1 | 4.24 | 22.2 | 0.6 | 3.75 | 203.27 | 14.07 |
| #2 | DBH2 | 14.2 | 4.29 | 16.0 | 0.6 | 3.21 | 116.95 | 14.16 |
| #3 | DBH2 | 13.1 | 3.38 | 15.5 | 0.6 | 3.17 | 110.91 | 10.35 |
| #4 | DBH2 | 11.0 | 2.68 | 11.4 | 0.6 | 2.86 | 66.00 | 8.13 |
| #5 | DBH3 | 5.5 | 2.01 | 6.2 | 0.3 | 2.05 | 22.86 | 6.34 |
| #6 | DBH3 | 5.3 | 1.60 | 5.0 | 0.3 | 1.45 | 13.44 | 5.50 |
| #7 | DBH3 | 4.2 | 2.26 | 5.1 | 0.3 | 1.50 | 14.14 | 6.91 |
| #8 | DBH4 | 3.8 | 2.11 | 4.1 | 0.3 | 1.00 | 7.85 | 6.55 |

Note: For the comparison of the sap flow density between trees, we grouped sampling trees according to the diameter at breast height (DBH1 > 20 cm, 20 cm ≥ DBH2 > 10 cm, 10 cm ≥ DBH3 > 5 cm, 5 cm > DBH4).

*2.4. Sapwood Area Estimation*

A 6-mm-diameter increment borer was used to determine the sapwood thickness close to where the sensors were installed in August 2014. The sapwood depth was determined with a dye method, using perchloric acid (at 40%) and Fehling's solution [20]; 10 to 15 min after dyeing, the sapwood becomes lighter and the heartwood becomes a darker-colored area. With this method, the sapwood depth was accurately measured with calipers and the sapwood area was calculated.

Using the calculated sapwood area, we obtained a sigmoidal growth function between the DBH and sapwood area (*As*) for 84 sample trees (Figure 3). This function was used to estimate the sapwood area for all trees at this site:

$$As = \frac{831.13}{(1 + 57.29e^{-0.13DBH})} \qquad (1)$$

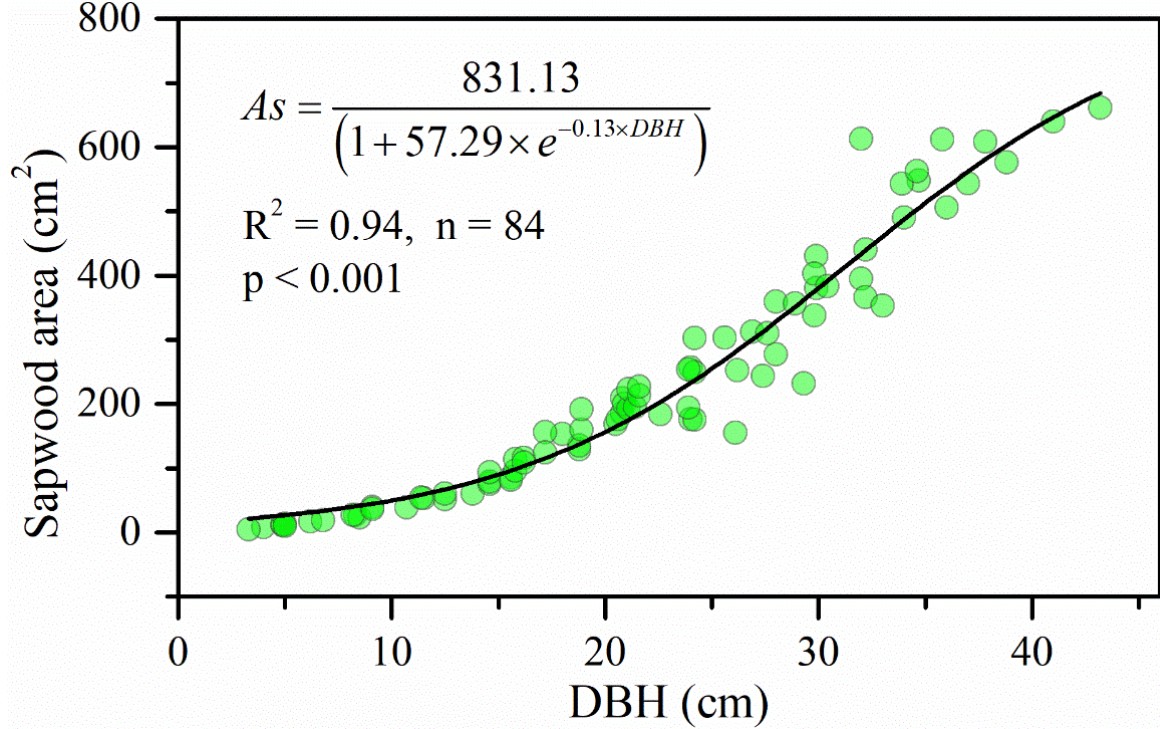

**Figure 3.** Relationship and fitting function between the diameter at breast height (*DBH*) and sapwood area (*As*) for Qinghai spruce (*P. crassifolia*).

*2.5. Sap Flow and Canopy Transpiration Evaluation*

Two thermocouples, named the outer point and inner point, were located at 12.5 and 27.5 mm from the epoxy base of each temperature-sensing probe of the HRM30 sensors. A point halfway between the two temperature probes was used to divide the sapwood into

the outer and inner annulus, and a single sap flow was calculated using the inner and outer sap flow velocities multiplied by the corresponding area.

Only a section of the xylem tissue contains sap, and in our study the sap flow velocity ($V_s$, mm h$^{-1}$) was calculated following the method used by Marshall [9] and modified by Barrett et al. [12]:

$$Vs = \frac{V_c \rho_b (c_w + m_c c_s)}{\rho s c s} \tag{2}$$

where $\rho_s$ (kg m$^{-3}$) and $\rho_b$ (kg m$^{-3}$) were the densities of the sample wood and water, respectively; $m_c$ (kg) was the water content of the sample sapwood; $c_w$ and $c_s$ were the specific heat capacities of the wood matrix, with about 1200 J kg$^{-1}$ °C$^{-1}$ and 4182 J kg$^{-1}$ °C$^{-1}$ of sap at 20 °C, respectively [21,22]; and $V_c$ was the corrected heat pulse velocity.

The sap flow ($Q$, kg h$^{-1}$ or L h$^{-1}$) was volumetric, calculated by multiplying the corrected sap flow velocity ($V_s$) by the cross-sectional area of the conducting sapwood ($A_s$):

$$Q = V_S A_S = V_{outer} A_1 + V_{inner} A_2 \tag{3}$$

where $V_{outer}$ and $V_{inner}$ were the corrected sap flow velocities of the outer and inner measuring points, respectively, and $A_1$, and $A_2$ were the corresponding annulus areas.

The sap flux density ($J_s$, kg m$^{-2}$ h$^{-1}$) was the sap flux of the sapwood area at breast height. Assuming that the sap flow velocity on a given day was the same per unit of sapwood area, $J_s$ was estimated as:

$$Js = \frac{Q}{A_1 + A_2} = \frac{Q}{A_S} \tag{4}$$

where $A_s$ is the sapwood area at the DBH and the canopy transpiration ($E_c$, mm d$^{-1}$) is the sap flux scaled up from tree-level and weighted by the canopy's projected area $A_c$ [18]. The crown projection area $A_c$ was estimated from below the crown by sighting it vertically at eight azimuths around each tree. The distance from the stem to each outermost projected point was measured and each cone area was calculated to estimate the crown projection areas (Table 1).

### 2.6. Sap Flux Variability and "Truing" Value Definition

To observe the changes over multiple growing seasons, we installed the sensors on sample trees between the 10th and 25th of April and removed them between the 15th and 25th of October each year. The probe installation was divided into two groups; the treatment group consisted of trees with probes installed at the previous insertion points after pre-filling with resin to ensure smooth re-insertion, and the control group consisted of trees with probes installed at a new insertion point close to the original wound.

To demonstrate the correctness of the two experimental groups, we analyzed the timing of the sap flux activity during the date available for analysis (DFA). Before the analysis, we determined the majority of environmental factors highly related to sap flow, and then quantitatively determined the DFAs. The response time of the tree physiological activity to the DFAs was measured for three periods of daily sap flux activity, namely the initial time (IOT), the cessation time (COT), and the end of the peak time (EPT).

Our observations were divided into three phases to address the sap flux characteristics. Phase 1 was a ten day intraday sap flow density process comparison during similar weather conditions between the 13 and 22 July 2015; all records were obtained at a half-hour scale to evaluate the sap flow records versus the physiological activity of the trees. Phase 2 addressed the relationship between the sap flow processes and environmental factors in the two growing seasons at a daily scale. Based on the results of the first two phases, the "truing" sap flux will be determined. In phase 3, we aimed to obtain the correction function and quantity of the sap flux density in the treatment group. Here, we implemented three repeated tests at three of the sampling trees between the 1st and 5th of August in 2015, with each tree having two sets of instruments as the treatment and control group, respectively.

All measurements were recorded at 30 min intervals and the details of the experimental design can be found in Table 2.

**Table 2.** Experimental design during the observation periods.

|  | Treatment Group | Control Group | Objective |
|---|---|---|---|
| Phase 1 | 13–17 July 2015 | 18–22 July 2015 | Physiological comparison daytime |
| Phase 2 | 18 July–15 September 2014 | 18 July–15 September 2015 | Driving factors analysis |
| Phase 3 | 1–5 August 2015 | 1–5 August 2015 | Sap flow quantification |

## 3. Results

### 3.1. Sapwood Area ($A_s$)

Based on 84 sampling records, we obtained a significant ($p < 0.001$) exponential function between the tree diameter at breast height (*DBH*, cm) and the sapwood area (*As*, cm$^2$) of Qinghai spruce (Figure 3). The evaluation function from Chang et al. (2014b) [17] did not include DBH < 5 cm and DBH > 35 cm, increasing the uncertainty of the fitting. For example, using the DBH of the eight sampling trees in this study and Chang et al.'s (2014b) [17] function, $A_s$ was overestimated by an average of 16.2% when the DBH > 35 cm, and underestimated by nearly 12.5% when the DBH < 5 cm.

### 3.2. Daily Sap Flux Characteristics during Two Growing Seasons

The daily sap flux density and canopy transpiration during the two growing seasons exhibited significant differences in both the quantity and magnitude over the two observation periods (Figure 4). The average value of the sap flux density was 50.7 kg m$^{-2}$ h$^{-1}$ in the treatment group and 83.83 kg m$^{-2}$ h$^{-1}$ in the control group, with similar magnitudes of coefficients of variation (12.24 kg m$^{-2}$ h$^{-1}$ and 35.7 kg m$^{-2}$ h$^{-1}$, respectively, over the two observation periods). The average canopy transpiration rates were $0.91 \pm 0.21$ and $1.38 \pm 0.55$ mm in the treatment and control group, respectively. The significant difference in the sap flux density and canopy transpiration between the treatment and control groups suggested that there were noticeable uncertainties in the treatment group or the control group measurements.

### 3.3. Diurnal Response Time of the Sap Flux Process

The differences in daily sap flow between the treatment and control groups indicated the existence of uncertainty; however, the differences could not validate the correctness or accuracy of the two groups. The results of correlation analysis (Spearman's rho) showed that the VPD, average temperature, and global short-wave radiation were significantly related to the daily sap flux density (Table 3), and the differences between the two seasons were marginal and not significant. The VPD was the most important factor in the treatment group, explaining nearly 90% of the variability in sap flow. The global short-wave radiation was the dominant factor in the control group, explaining nearly 88% of sap flow variability.

Based on the characteristics of the three major meteorological factors (VPD, air temperature, and global short-wave radiation), we selected the DFAs and divided them into sunny days (day of the year, DOY 194 and 200) and rainy days (DOY 197 and 202) to include various weather conditions in the analysis (Table 4). The response times exhibited no differences among the three DBHs; the treatment group exhibited no clear activity during the day, while the control group exhibited the IOT at 07:00, COT at around 18:30, and EPT at 23:00. The sap flux ceased at around 13:30 in the control group and the leaf stomata closed; however, no cessation signal was observed in the treatment groups on sunny or rainy days (Figure 5).

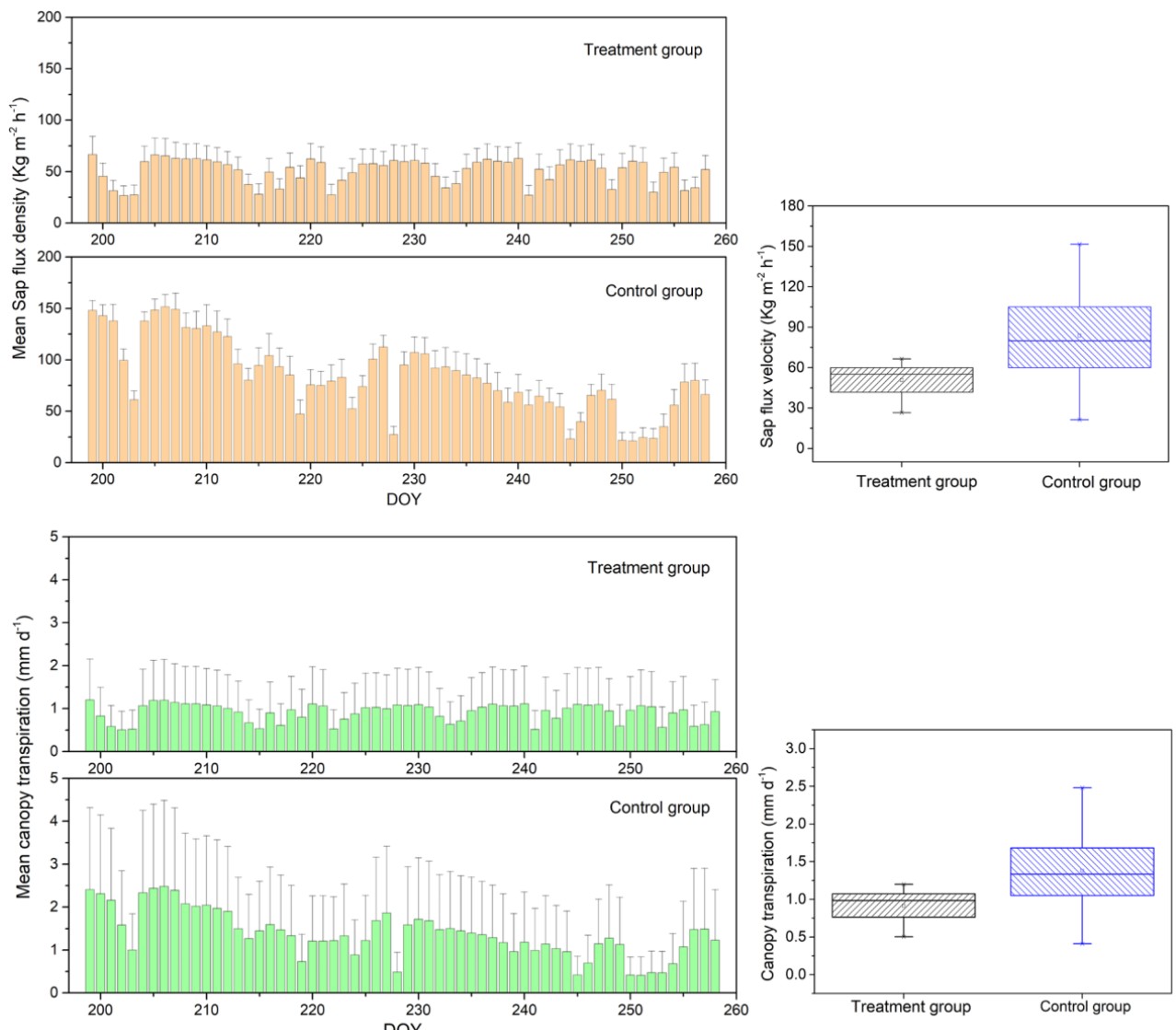

**Figure 4.** Daily sap flux density and canopy transpiration rates during the growing season for treatment and control groups (daily mean and standard deviations), and box plots of values averaged for the season.

**Table 3.** Spearman's rho correlations between the sap flux density, canopy transpiration, and environmental factors.

| Variable | Evaluation Indexes | VPD | Soil Water Content | Average Temperature | Global Short-Wave Radiation | Mean TWD |
|---|---|---|---|---|---|---|
| Daily sap flux density in 2014 | Correlation coefficient | 0.900 ** | −0.081 | 0.738 ** | 0.769 ** | 0.557 ** |
| | Sig. | 0.000 | 0.538 | 0.000 | 0.000 | 0.000 |
| Daily canopy transpiration in 2014 | Correlation coefficient | 0.896 ** | −0.075 | 0.742 ** | 0.778 ** | 0.543 ** |
| | Sig. | 0.000 | 0.571 | 0.000 | 0.000 | 0.000 |
| Daily sap flux density in 2015 | Pearson correlation | 0.706 ** | 0.037 | 0.704 ** | 0.870 ** | −0.181 |
| | Sig. | 0.000 | 0.779 | 0.000 | 0.000 | 0.165 |
| Daily canopy transpiration in 2015 | Pearson correlation | 0.696 ** | 0.089 | 0.662 ** | 0.885 ** | −0.179 |
| | Sig. | 0.000 | 0.498 | 0.000 | 0.000 | 0.170 |

Notes: ** indicates significance at the 0.01 level (2-tailed).

**Table 4.** Days selected for the comparison analysis of the treatment group and the control group.

| DOY | Date | Precipitation (mm) | VPD (Kpa) | Average Temperature (°C) | Global Short-Wave Radiation (W m⁻²) |
|-----|------|--------------------|-----------|--------------------------|-------------------------------------|
| 194 | 13 July 2015 | 0 | 0.37 | 12.4 | 318.8 |
| 200 | 19 July 2015 | 0 | 0.36 | 13.7 | 326.6 |
| 197 | 16 July 2015 | 4.5 | 0.24 | 10.0 | 316.3 |
| 202 | 21 July 2015 | 5.0 | 0.26 | 12.0 | 308.4 |

Note: DOY means day of the year; VPD means vapor pressure deficit.

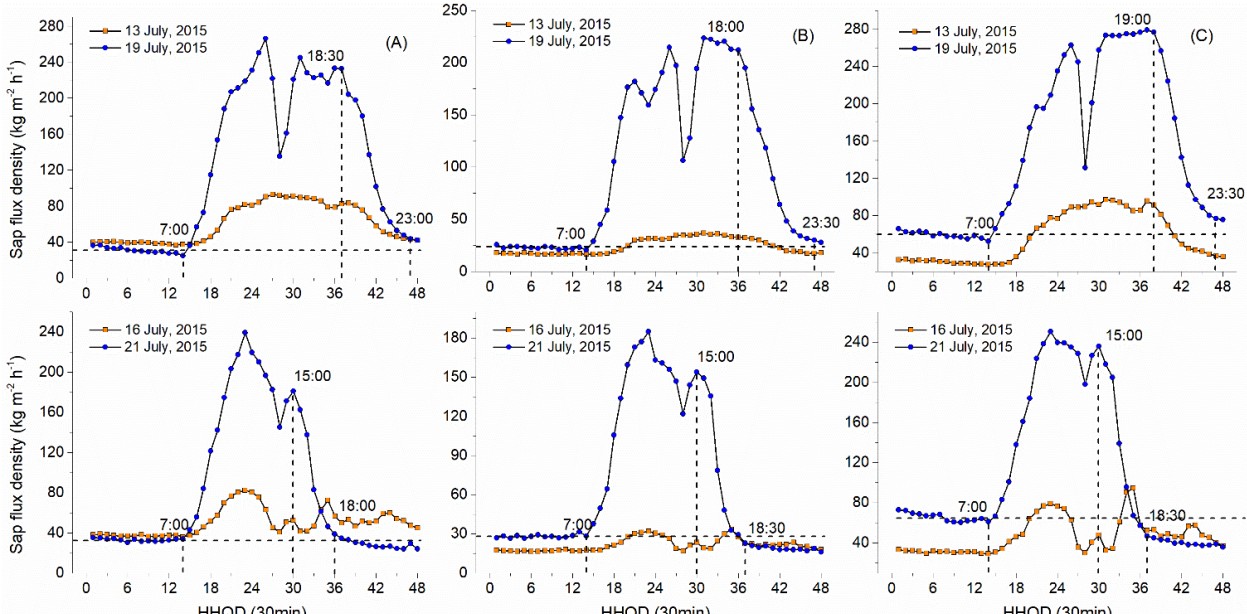

**Figure 5.** A daily course of sap flux density activity for 2 days with similar weather conditions during July 2015. The DBHs of columns (**A**–**C**) were DBH2, DBH3, and DBH4, respectively.

*3.4. Sap Flux Analysis and Simulation Evaluation*

The relationship between the sap flux density and meteorological factors is also an important and robust criterion for a 'truing' value analysis for long timescales. This relationship was different for the treatment (Figure 6) and control groups (Figure 7) in this study, with a power function in the treatment group and a linear function in the control group. The coefficient of determination also differed for the four meteorological indexes, and strong relationships ($R^2 > 0.85$) were observed for global short-wave radiation and VPD in the treatment group ($R^2 > 0.75$) and for global short-wave radiation in the control group, while the VPD and average temperature exhibited a relatively weak correlation. The *P* values of the six regression models were < 0.001. Finally, it is worth noting that there was no relationship between the TWD and sap flux density for either group.

The comparison of the two sap flux measurement methods showed that firstly the sap flux processes were the same for both groups, secondly the sap flux density value in the treatment group was lower than that in the control group throughout the test period, and finally the mornings on three of the five test days exhibited a distinct sap flow disturbance in both groups (Figure 8).

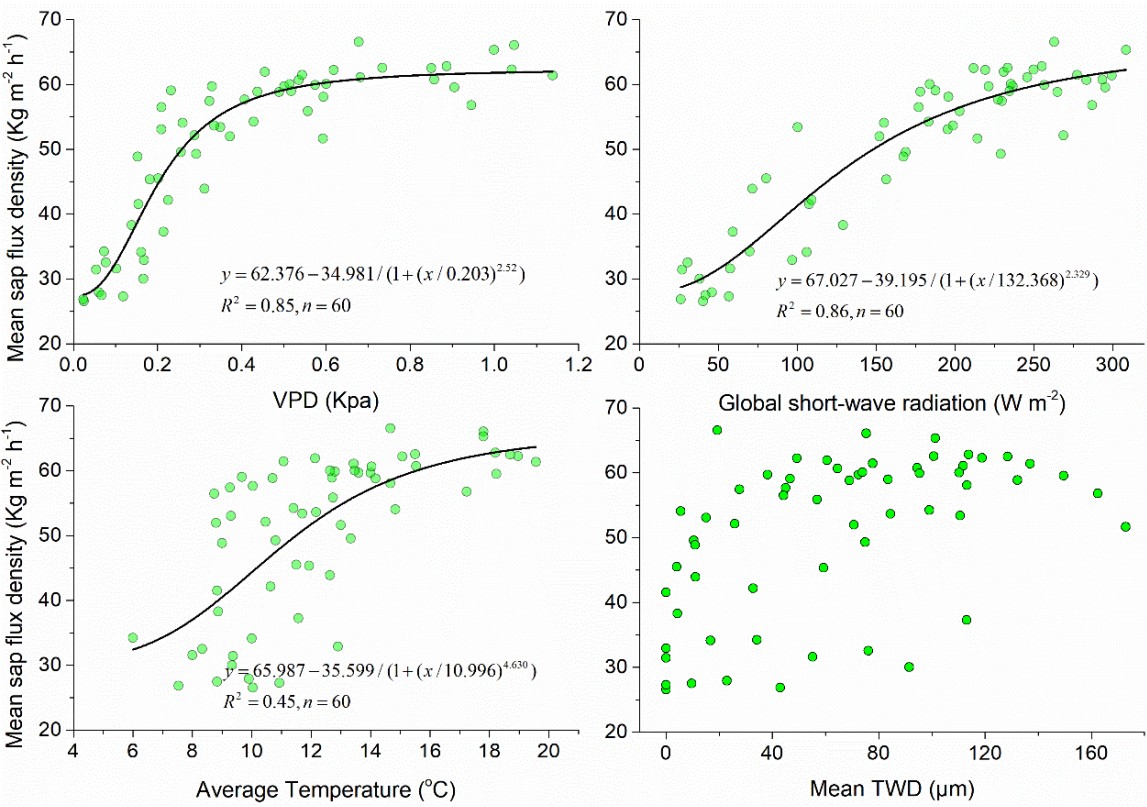

**Figure 6.** Relationships between sap flux density and vapor pressure deficit (VPD), global short-wave radiation, average air temperature, and daily tree water deficit (TWD) for the treatment group over the growing season.

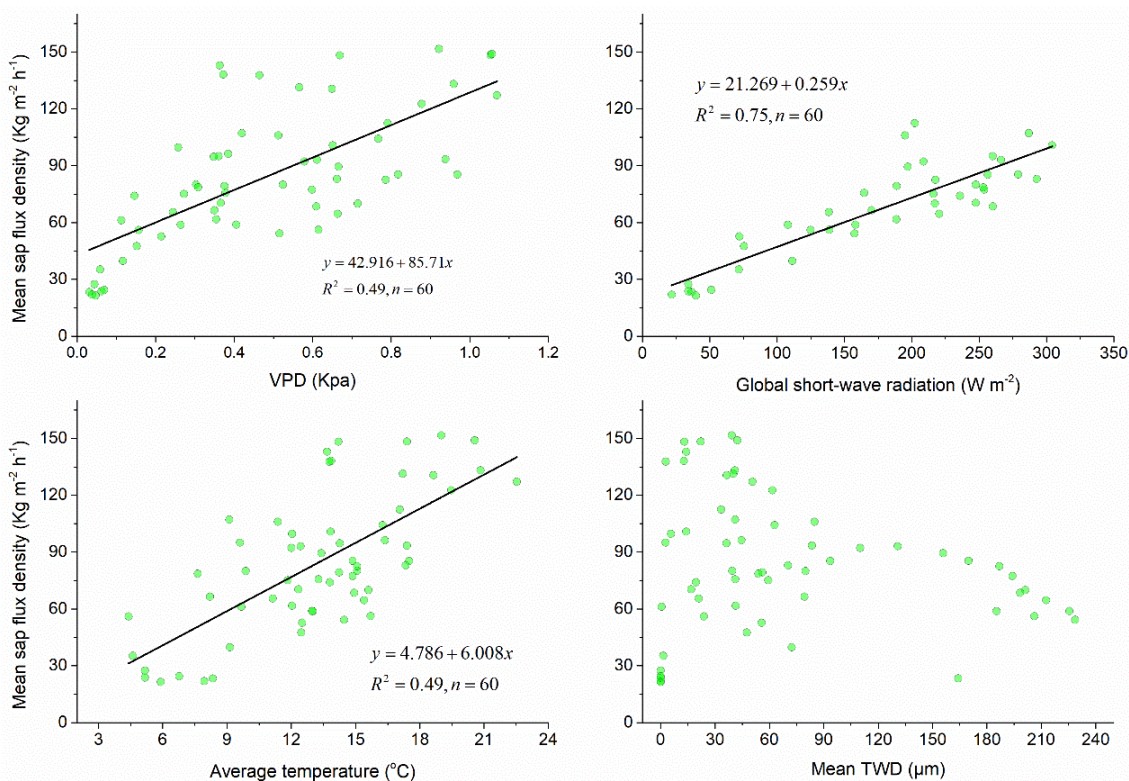

**Figure 7.** Relationships between sap flux density and VPD, global short-wave radiation, average air temperature, and daily TWD for the control group over the growing season.

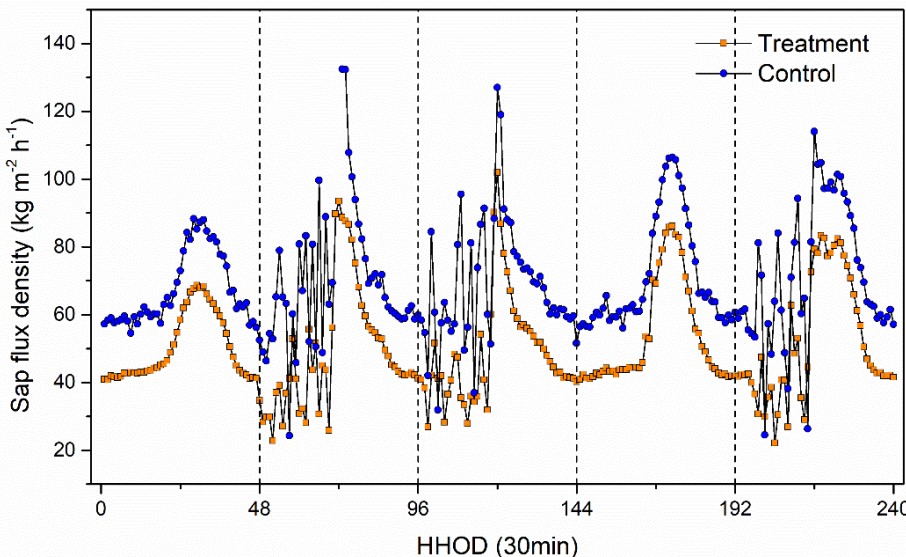

**Figure 8.** Sap flux density comparison for the treatment and control groups during five observation days in 2015. HHOD means half-hour-of-the-day.

To quantify the difference between the treatment and control groups, we fitted sap flux density records at a 30-min scale (Figure 9) and calculated the average sap flux density and summed canopy transpiration values (Table 5). We found a significant relationship ($p < 0.001$) between the treatment and control group for both the mean sap flux density and canopy transpiration, whereby the mean sap flux density and canopy transpiration in the treatment group were underestimated by about 37% compared to the control group.

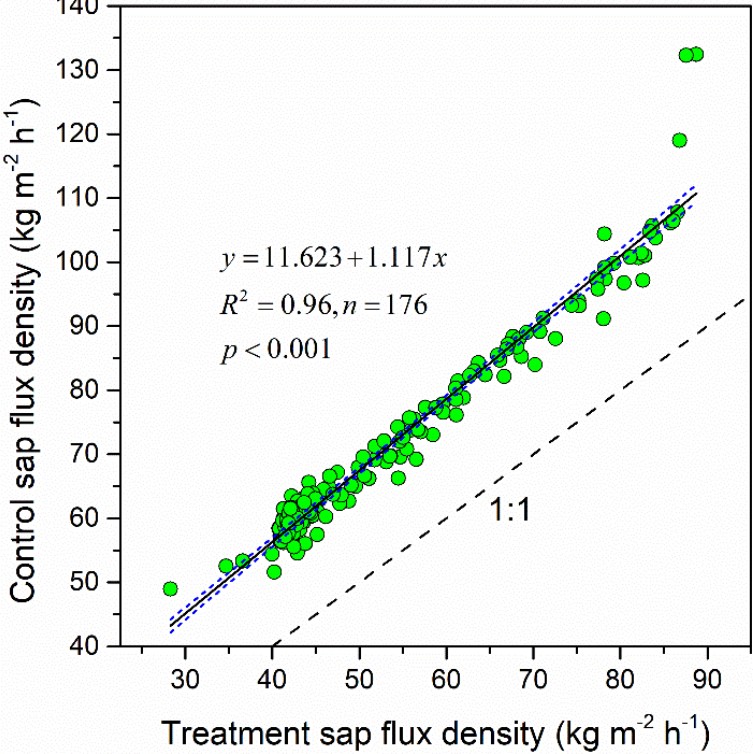

**Figure 9.** Relationship (black solid line) between the treatment and control groups in terms of the sap flux density, where the blue upper and lower dashed lines are the confidence intervals for the 95% level; the black dashed line is the 1:1 reference line.

**Table 5.** Statistical results of two indexes between the treatment group and the control group.

| Variable | Mean Sap Flux Density (kg m$^{-2}$ h$^{-1}$) | Mean Canopy Transpiration (mm d$^{-1}$) |
|---|---|---|
| Treatment group | $50.80 \pm 12.24$ | $0.44 \pm 0.01$ |
| Control group | $69.82 \pm 35.70$ | $0.60 \pm 0.55$ |
| Difference | 37.5% | 36.8% |

## 4. Discussion

### 4.1. Influence of DBH Sampling on the Sapwood Area Estimation

Using the DBH to estimate the sapwood area is one of the principal non-destructive sap flow estimation techniques [23]. Previous studies have shown that much of the error is related to the measurement of the sapwood area [24,25]. The other sources of error are due to using a single point in the sapwood calculation, inaccurate estimation of the variability in sap flow at different depths within the sapwood [24], non-uniform distribution of the sapwood vessels [15], and variations in the radial profile within the sapwood [26]. Additionally, the fitting function may be used while its accuracy is ignored, and an incomplete and uneven sampling scheme may lead to an inaccurate fitting function and incorrect sap flow estimation. Our study is one of the first to address the variability in the sapwood area estimation function in conifer forests.

In this study, incorporating a range of DBHs significantly increased the accuracy of the sapwood estimation. Chang et al. [18] overestimated the sapwood area for large DBHs and underestimated it for small DBHs; in particular, when DBH = 11.4 cm, $A_s$ was overestimated by > 28.5%, and when DBH = 4.1 cm $A_s$ was underestimated by > 22.6%. These results suggest that when $A_s$ was estimated with a mean DBH of 9.77 cm, the function used by Chang et al. produced a significant level of uncertainty, whereby insufficient records resulted in significant estimation errors. Bodo and Arain [24] used 17 tree measurements for the sapwood area estimation from the DBH, and their results and conclusions should be treated with caution. Furthermore, Looker et al. [25] stressed that $A_s$ affected not only the sap flow calculations but also the conversion of the heat pulse velocity to the sap flux density. Thus, the uncertainty in the sapwood area fitting function may have far-reaching implications.

### 4.2. Relationships between Environmental Factors and Sap Flux Density at Various Time Scales

Sap flow estimation is one of the prominent methods used in the determination of whole-tree water use and the monitoring of tree physiological activity in forest management [4]. The sap flux gradually intensifies and the physiological activity increases as the sun rises, then suddenly decreases when the leaf stomata close around noon; thus, the sap flux activities can reflect tree physiological processes during a day [27,28].

Based on this, a detailed account of the sap flux density progression for one day (Figure 5) confirmed that the control group in this study more closely reflected actual processes compared with the treatment group under similar weather conditions; this indicated that the treatment method exhibited an inaccurate estimation for the processes of the sap flux density, possibly due to repeated drilling in the wound. Steppe et al. [29] and Burgess and Downy [15] observed a severely reduced magnitude and dampened diurnal pattern of sap flux using heat pulse velocity measurements in sample trees. However, other studies showed [13,14,29] that the wounding coefficient can be used to correct a range of wound sizes. In this study, the drilling holes filled with solidified resin after several growing seasons, and variations in material composition were observed between tree xylem samples. Thus, precautions should be taken and new correction factors and functions should be explored.

The sap flux was not only affected by wounding but also by meteorological factors such as the VPD, global short-wave radiation, average temperature, and TWD [17,25,30]. In 2014, Chang et al. [17] showed a linear correlation between the global short-wave

radiation (less than 800 W m$^{-2}$) and sap flux density, which was highly consistent with the results for the control group in this study (Figure 7); in this study, the global short-wave radiation explained 75% of the variability in sap flux density, followed by the vapor pressure deficit and air temperature, also consistent with previous studies [31]. This indicated that the treatment method could not capture the pattern of sap flux fluctuation during the growing season.

However, the fitting function of the VPD and air temperature varied in this study, possibly due to the limited value range of the available records. For instance, when the VPD was < 1.2 Kpa, the relationship was close to linear, but it varied over a wider data range [31]; the air temperature exhibited a similar response. This exposed the uncertainty of the relationship obtained with limited data, and we propose that a wider range of records will increase the accuracy of this relationship. Additionally, using the half-hour time scale rather than the average daily values may increase the accuracy in analyses of the relationships among variables.

*4.3. Sap Flow Evaluation and Estimation*

Many studies have focused on correction factors for determining the stand transpiration using sap flow measurements [17,32–34]. One of the aims of this study was to determine the actual difference in treatment groups and whether or not we can determine the correction function for young Qinghai spruce sap flow estimates. The results of the five day sap flux density process in the same tree at DBH with two adjacent positions on the north side of the tree trunk with two methods are shown in Figure 8. The relationship between the two groups was significantly linear, with some noise, especially between 00:00 and 12:00 on three of the five days, indicating the well-known limitation of heat pulse velocity sensors [13,21], although this may not be applicable in this study. However, the limited observations in this study limit our ability to explain this.

Furthermore, to quantify the difference between the two groups, the sap flux density and canopy transpiration were summarized using SPSS. Both groups exhibited high linearity and precision (Figure 9), with the regression line being almost parallel to the 1:1 reference line and showing a strong correlation ($p < 0.001$). This result indicated that we could estimate the sap flow from the treatment group to minimize the observation damage to the young trees in future work.

**5. Conclusions**

We compared two sap flow sensor installation methods during several growing seasons in Qinghai spruce. Our results revealed uncertainties and differences in the two methods, especially for young trees (DBH < 5 cm). The results indicated that the representativeness of the tree DBH measurements is critical for accurate sapwood area estimations. Although sap flow measurements using a previous drilling hole exhibited several uncertainties during the growing season, a close-fitting linear regression was obtained for the data correction and the results satisfied a five-day observation period, suggesting that the tree damage can be minimized during sap flow observations with the HRM method.

**Author Contributions:** Conceptualization, Z.H. and J.Y. (Junjun Yang); methodology, J.Y. (Junjun Yang); software, J.Y. (Junjun Yang) and P.L.; validation, Q.T., J.D. and P.L.; investigation, J.Y. (Junjun Yang) and P.L.; resources, Z.H. and J.F.; data curation, J.Y. (Junjun Yang),W.Z., L.G. and Y.L.; writing—original draft preparation, J.Y. (Junjun Yang); writing—review and editing, J.Y. (Junjun Yang), G.W. and J.Y. (Jialiang Yan); visualization, J.Y. (Junjun Yang) and W.Z.; supervision, J.F. and L.G.; project administration, Z.H.; funding acquisition, J.Y. (Junjun Yang) and Z.H. All authors have read and agreed to the published version of the manuscript.

**Funding:** This study was funded by the National Natural Sciences Foundation of China (41901050, 41901044, and 41621001); Strategic Priority Program of the Chinese Academy of Sciences, Grant or Award Number: XDA23060301; and Key Program of the Chinese Academy of Sciences, Grant or Award Number: QYZDJ-SSW-DQC040.

**Institutional Review Board Statement:** Not applicable.

**Informed Consent Statement:** Not applicable.

**Data Availability Statement:** Not applicable.

**Acknowledgments:** We would like to thank Kathryn Piatek for her suggestions and editorial assistance in writing this article. We thank the two reviewers for their helpful contributions in clarifying the text.

**Conflicts of Interest:** The authors declare no conflict of interest.

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
