# Peer review of "Variability in Minimal-Damage Sap Flow Observations and Whole-Tree Transpiration Estimates in a Coniferous Forest"

_water, doi:10.3390/w14162551_

Round 1

Reviewer 1 Report

Suggested changes to the wording of the article:

 Abstract

Line 24: …The results showed that an incomplete diameter at breast height (DBH) range …

Introduction

Line 42: …from differences in the methods

Line 54: … resolve a range by sap flux…

Materials and methods

Line 154 to 161:

Using the calculated sapwood area, we obtained as a sigmoidal growth function between DBH and sapwood area (As) for 84 sample trees (Fig. 3). This function was used to estimate sapwood area for all trees at this site.

In function (1) use DBH and not D. It gets confusing as performed by the authors.

 Figure 3. Relationship and the fitting function between diameter at breast height (DBH) and sapwood area (As) for Qinghai spruce (P. crassifolia).

 Results

Line 222: …tree diameter at breast height (DBH, cm) and sapwood area (…

Line 254: …days (day of the year, DOY 194 and 200) and….

The sections (Introduction, Materials and Methods, Results, Discussion, and Conclusions) and their subsections are not in correct increasing numerical order throughout the manuscript.

Author Response

Dear Editorial Office,

Thank you for your letter and for the reviewers’ comments on our manuscript titled “Variability in minimal-damage sap flow observations and whole-tree transpiration estimates in a coniferous forest” (1860469). The comments and suggestions were invaluable in revising and improving our manuscript. We have addressed the comments carefully and made corrections. As advised, we revised the manuscript according to the two reviewers and the figure file marked up using “Track Changes”; named “water-1860496(0815).docx” has been uploaded to the Editorial Manager. We hope the revised manuscript meets with your full approval.

Sincerely,

Junjun Yang, Zhibin He, Pengfei Lin, Jun Du, Quanyan Tian, Jianmin Feng, Yufeng Liu, Lingxia Guo, Guohua Wang, Jialiang Yan and Weijun Zhao

Reviewer 2 Report

Transpiration from forest regions is an important component of the water cycle, but evapotranspiration (ET) measurements or estimates are difficult to obtain and their quality varies with both measurement method and conditions.

The authors addressed a relatively simple question— can the same holes in trees be reused for sap flow measurements or does this procedure introduce errors? They present the results of a simple experiment using control (new holes) and treatment (old holes) measurements of sap flow.  They did very careful and complete measurements and were able to demonstrate significant differences in sap flow rates between the treatment and controls. These differences affected the relationships between sap flow rates and environmental variables (vapor pressure deficit, etc.)

From their other measurements, they were also able to demonstrate the importance of measurements of tree diameter and sap wood thickness to obtain accurate measurements of sap flow and transpiration for forests composed of trees of varying sizes.  

This is an excellent paper.

Author Response

(The authors gave the same response as above.)
